# Colored Heirloom Corn as a Public Good: The Case of Tlaxcala, Mexico

Marisol Velázquez-Salazar [1],*, Germán Scalzo [1] and Carmen Byker Shanks [2]

1 Facultad de Ciencias Económicas y Empresariales, Universidad Panamericana, Augusto Rodin 498, Ciudad de México 03920, Mexico; gscalzo@up.edu.mx
2 Department of Health and Human Development, Montana State University, Bozeman, MT 59717, USA; cbykershanks@montana.edu
* Correspondence: mvelazquez@up.edu.mx

**Abstract:** Valorization of territories with diverse cultures and heritage has multiplied in recent years. This study analyzes the case of colored heirloom corn in Tlaxcala, Mexico, as a potential public good associated with the region's biocultural heritage. The analysis conducted herein relies on a wide range of literature from relevant theory, including Geographical Indications, Global Value Chains, Community-Based Entrepreneurship, Public Goods, and Sustainable Development, in order to employ case study methodology. We leverage a novel approach to analyze the heirloom corn chain and its publicness. This chain reveals its status as a potential public good that clearly influences biocultural heritage, which has been preserved by several generations. To preserve colored heirloom corn in Tlaxcala, Mexico, a development strategy is needed that links actors and resources, involves the public sector, and furthers expansion of the private sector.

**Keywords:** biodiversity; colored heirloom corn; community-based enterprise; public goods; sustainable development; value chain

## 1. Introduction

Latin America is a powerhouse of genetic, natural, and cultural biodiversity [1]. Cultural diversity highlights the identity of a place, formed from a narrative of distinctive and plural traditions and stories that comprise a cultural heritage. Valorization of territories with diverse cultures and heritage has multiplied in recent years, especially in rural areas of developing countries [2]. Frequently, these cases represent the relationship that local people have with their natural environment, or their biocultural heritage [3]. Due to their endogenous nature, these initiatives are presented as a more appropriate alternative to the hegemonic development models that fail to contemplate local particularities [4].

This study analyzes colored heirloom corn in Tlaxcala, Mexico as a potential public good associated with the region's biocultural heritage. Heirloom corn refers to corn that is considered part of native corn crops; it dates back 6500 years with teocintle, which is thought to be the parent plant of corn that is still produced in Mexico to this day.

To do so, our research is framed within Belletti, Marescotti, and Touzard's [5] (2017) Geographical Indications, Public Goods, and Sustainable Development proposal. Likewise, the corn chain is explained using Gereffi's Global Value Chains approach [6,7] since it reveals the publicness of this specific product based on the dimensions associated with the value chain. Conceptual and methodological elements regarding territory, biodiversity, community-based enterprise, and development are applied to connect theory with practice in the field.

Using the aforementioned theoretical framework, the analysis conducted herein employs a case study methodology given that the specific case of heirloom corn requires attention that is different from the attention that white, yellow or other corn for human consumption usually receives. This difference lies in considering heirloom corn a piece

of biocultural heritage in the Tlaxcala area, taking into account that this consideration involves transferring real monetary value from said heritage to the product.

## 2. Theoretical Framework

### 2.1. Public Goods and Publicness

This study focuses on the theoretical framework of public goods as part of biocultural heritage. For this, Belletti, Marescotti, and Touzard's [5] approach is used, establishing that public goods are social constructions determined by conventions, collective actions, and public policies. This approach goes beyond considerations of consumption alone to understand that publicness must also be present in the production and distribution process. That is, the duties and costs—as well as profits—are socialized within and sometimes outside of the territory. With an economic logic, Samuelson [8,9] defines a public good as one that, once produced, can be consumed by additional consumers at no additional cost. It is non-rival and non-excludable, and almost always produced by the government. As can be seen, this definition limits the concept to goods generally produced by governments and specifically public services for citizens' use.

Belletti, Marescotti, and Touzard [5] complement Samuelson's definition in two ways. They do so first by expanding it towards the sphere of production and away from a sole focus on distribution and consumption; second, they define publicness beyond government provisions, extending it to goods whose production implies a common benefit in terms of biocultural heritage. The former implies that producers of the primary link—especially in the agri-food sector—benefit from inherited heritage in cultural and environmental terms within the territory and from the knowledge of the productive practices that have preserved said assets. Within production, publicness is presented in collective decision-making. The community of actors surrounding a public good socializes decision-making based on discussion and local participation. The latter implies that a public good can be produced, transformed, and distributed by local actors collectively, with common benefit and is not exclusive to the government. These goods are preserved and conserved by various levels of actors, from local to external and from formal to informal. They are objects of common interest in the territory, and are also recognized by a wide spectrum outside of it. Likewise, they are meant to be appropriated collectively rather than individually.

According to Belletti et al. [5], there are five types of publicness depending on the level of impact and connection with other actors and resources. First, there are common goods that influence other public goods or resources, such as the ecosystem, traditional knowledge, agricultural systems, or local gastronomy. Second, some public goods impact the socio-economic environment of the territory in terms of employment, income, and social cohesion. The third refers to a Geographical Indication (henceforth GI) as a territorial public good whose reputation offers relevant opportunities for the local population. The fourth level points to a GI as a common good in the value chain where there is a set of rules that are formalized as GI or Denomination of Origin (henceforth DO), combating free riding, aligning individual interests in collective action, reducing transaction costs, and achieving economies of scope and scale. The fifth type pertains to patrimonial public goods, which become so because of the biological, symbolic, and cultural meanings that transcend the territory in which they are produced and that have achieved recognition as valuable in themselves. These assets give a territory an identity based on the social construction around an asset or resource.

Public goods, based on their plurality and social construction, are fragile and vulnerable to appropriation within the region because anyone can make use of them or produce them, thus they can be easily exploited or wasted according to individual interests if there is no collective organization or institution to protect them. Overexploitation that ignores preservation is common, as are cases of under-exploitation when the resource ceases to be profitable and is abandoned. Both of these paths lead to the disappearance or loss of the common good in question.

Following Belletti et al. [5], two conditions must be met to activate the sustainable development of public goods in rural areas. The first pertains to a strategy that values resources by connecting stakeholders at different levels, including local actors, institutions, and public policies; the second to establishing formal and informal collective rules for the coordination, management, and use of common resources. When one or both are absent, the public good tends to get lost in the market because its intrinsic value—based on local production under a traditional system—fades.

A public good is, then, a resource that connects to other resources within a territory; it is characterized by collective action over time and is, or has the potential to become, part of the territory's biocultural heritage.

### 2.2. Value Chains (VC)

The value chain approach reveals the chain's general structure, from inputs to consumption; it also spatially locates productive processes and spaces for consumption, makes visible economic power relations between the links and throughout the chain, analyzes the norms and conventions, and evaluates the role of institutions with respect to the product [6,7,10–12]. Capturing the dimensions associated with input–output, territoriality and the institutional framework, and coupling the knowledge of how relevant governance entities establish the meaning of domain, connection, and regulations, reveals a complete image of the value chain which allows us to measure the publicness potential of the good.

According to Velázquez [13], a value chain's behavior can be described using the four dimensions that Gereffi [6,7] proposed where each one is subdivided into categories, variables, or observables, which are then organized into different indicators. Table 1 includes the proposal originally based on global value chains, adds the conceptual framework of public goods to establish the link between both methodologies and, in so doing, explains how the level of publicness can be identified from the global value chains approach (See Table 1). The input–output dimension refers to the sequence of added value and is subdivided into product characteristics and income distribution among the participating actors and the chain's structure. The spatial or territorial dimension refers to production concentration and distribution networks, and is subdivided into topics surrounding economic geography, like the socioeconomic characteristics of producers and consumers, as well as the geography of exports. This dimension involves knowledge of the territory, from its socioeconomic and material aspects, to its immaterial elements that include landscapes, tradition, and culture. From this perspective, the public good is analyzed as part of the territory's biocultural heritage. The institutional framework, for its part, is the context under which the value chain of the potential public good operates; its importance is found in the fact that the norms and rules established therein, through public or private, government or non-governmental organizations, promote or marginalize certain producers' entry into value chains, as well as contribute to the strength or vulnerability of the public good. Food security, defined by The Food Agriculture Organization (FAO) as "when all people, at all times, have physical, social and economic access to sufficient, safe and nutritious food that meets their dietary needs and food preferences for an active and healthy life" [14], is included in the institutional framework dimension. Food insecurity and hunger remain a significant concern globally and combatting them is part of why the good is included in this dimension, namely because of the importance that the good has in terms of national and local consumption.



**Table 1.** Dimensions, categories and indicators for evaluating the heirloom corn chain and its publicness.

| Dimension | Category or Variable | Indicators for the Heirloom Corn Chain | Link with Public Goods |
|---|---|---|---|
| **Input-Output** | Characteristics of the marketed product | Race, variety, color, nutritional properties | Product differentiation |
| | Income distribution along the chain | Final consumer price<br>Unit value | Collective benefit from local participation |
| | | Planting, harvesting, fertilization, pest control, land preparation, interests<br>Price obtained by the producer<br>Yield per hectare<br>Total income per hectare | |
| | GVC structure | Definition of participants in each link | Connection of actors with other actors within the chain |
| **Economic Geography** | Geo-economic structure of production | Corn production | Knowledge of the territory, economics, traditions, history, culture, society, biocultural heritage |
| | | Socio-economic characteristics of primary maize producers (type of property, place of production, marginalization stratum, human development index) | |
| | Geo-economic structure of consumption | Consumption of maize products<br>Establishments for consumption of maize and other by-products<br>Features of end consumers | |
| **Institutional Framework** | Institutional context | National and international bodies and institutions and their main maize roles and supports | Participation of external actors: public, non-governmental organizations and private actors external to the chain |
| | Protection and production support laws | Food security<br>Seeds and transgenic maize laws<br>Federal Law for the Promotion and Protection of Native Corn<br>Heirloom Corn Protection and Conservation Law | |

Source: Authors' elaboration based on [5–7,15,16].

The fourth dimension is the most complex since it tries to identify the actor (s) that dominates the value chain, their relationship with primary producers, the coordination between them and the codes that govern their interaction. Likewise, this dimension analyzes the conventions that apply along the chain and if the good's attributes and qualities are valued. The following table illustrates Velázquez's [15] typology proposal for the understanding of heirloom corn. Governance understood from the approach of global value chains as a domain of the chain, and as a link between actors and regulations, is widely related to the concept of public goods since it determines how economic power relations are set up within the chain and the influence that some actors have over others.

According to Table 2, there are up to five types of governance, each of which is dominated by the producer (B) or by the buyer (A), understood as the intermediary or retailer, and not as the final consumer. It determines the agent that exercises domain or power over the chain [6]. Agro-food chains are controlled, in general, by the buyer [13,16–19]. The chain's connection is related to asymmetric power according to the level of coordination and is the ability to provide and execute instructions; it is evaluated by observing the

interaction between transactions' complexity and coding ability, as well as providers' response capacity between the primary producer and the buyer, understood as a commercial company. Increased coordination is associated with higher levels of asymmetric power [17]. In terms of public goods, greater coordination means less independence for the actors and greater asymmetry, which usually translates into unequal income distributions for the final links. In this case, greater coordination is the opposite of working collectively.

**Table 2.** Typology of governance.

| | Governance | | | | | Indicators | | | | | |
|---|---|---|---|---|---|---|---|---|---|---|---|
| **TYPOLOGY** | **DOMAIN** | **LINKAGE** | **CONVENTION** | **MARKET** | **INTERMEDIARIES** | **COORDINATION LEVEL** | **ASYMMETRY** | **COMPLEXITY OF TRANSACTIONS** | **HABILITY IN CODING TRANSACTIONS** | **CAPACITY OF RESPONSE OF THE SUPPLIERS** |
| **G1A** | Buyer | Market | Market | Traditional market | Price | Extreme Low | Extreme Low | Low | High | High |
| **G2A** | Buyer | Modular | Industrial | | Key intermediary | Middle-low | Middle-low | High | High | High |
| **G3A** | Buyer | Relational | Domestic | Several producers, one buyer | Relational intermediary | Middle | Middle | High | Low | High |
| **G3B** | Producer | Relational | | | | Middle | Middle | High | Low | High |
| **G4A** | Buyer | Captive | Domestic, Industrial, Opinion | | Intermediaries | Middle-high | Middle-high | High | High | Low |
| **G4B** | Producer | Captive | | | | Middle-high | Middle-high | High | High | Low |
| **G5A** | Buyer | Hierarchical | Domestic, Industrial, Opinion | Monopoly or Oligopoly | Without intermediaries | High | High | High | Low | Low |
| **G5B** | Producer | Hierarchical | | | | High | High | High | Low | Low |

Source: Authors' elaboration based on [15].

There are five levels ranging from the conventional market in which the price determines supply and demand, to the captive level in which the producer must follow the buyer's instructions or risk being left out of the chain. The highest is the hierarchical level in which there are imperfectly competitive markets, monopolies or oligopolies, which require the greatest level of coordination and make public goods less likely to emerge. In the agro-food industry, in most cases, the company has a vertical structure and owns all the links in the chain, from control of the fields, transformation, and even commercialization [7]. This is why public goods initiatives in this sector are relevant for analysis.

The next category is governance through normativity; it indicates the type of convention that the chain presents, refers to the set of agreements translated into rules by social convention and sets the type of relationship that exists between the consumer and the product. The market-type convention explains the difference in price starting from different qualities; the domestic convention demonstrates that product choice is determined by long-term relationships and brand awareness. The industrial convention is present when quality is determined by means of norms or standards evaluated by a third party; the civic convention includes commitment to the collective well-being, where social or

environmental benefits are determining factors in the chain. The inspirational convention includes creativity, innovation, vision, uniqueness, and other elements considered artistic or artisanal that influence choice; finally, the opinion convention establishes that judgments from specialists matter at a subjective level in the chain [11].

This theoretical framework has been used for several years, dating back to its proposal in the 1990s. However, it has only recently become a benchmark for the study of chains as shown in the different applications and studies carried out by international organizations such as Food and Agriculture Organization (FAO), United Nations (UN) Environment, Economic Commission for Latin American and the Caribbean (ECLAC), and the World Bank, among others. Its justification focuses on the fact that it enables analysis and evaluation of chains starting from a set of interactions inside and outside of the chain, in addition to allowing for comparisons between regions, countries, products, and even sectors. Studies using this framework can be done at the micro level, examining it link by link, the meso level, including relationships between actors, and can even include the entire chain, providing a good approach to VC behavior and the level of public good. On the other hand, at a theoretical level it is possible to tie this framework in with others, given its flexibility and openness.

The Value Chain approach serves as an analytical tool for other approaches, which the public goods approach employs. This model's strength is found precisely in that it understands that each good is different and must be approached according to its own characteristics.

## 3. Methods and Materials

Analysis of the colored heirloom corn chain was carried out using the indicators presented in Table 1, with a mixed quantitative and qualitative method. Secondary data were obtained for the 2019–2020 period in Mexico and broken down by state; they were found at official sources, namely the National Commission for the Knowledge and Use of Biodiversity and the Bank of Mexico. We collected secondary data to find information about how the heirloom corn chain is promoted as a potential public good for biocultural heritage within Tlaxcala and Mexico more broadly.

Primary data was collected using an ethnographic approach. Authors visited Tlaxcala in 2019 to observe the setting and interview producers, and conducted another follow-up interview in June 2020 after that year's corn harvest and during the COVID-19 pandemic. The in-depth interviews with key actors used the snowball technique. Interviews were conducted with key actors based on their connection to other local producers and the quality and quantity of information they were able to provide. The data collection materials included technical files to gather basic information regarding interviewees' socioeconomic conditions, forms of production, consumption, and self-consumption. These heirloom corn producers were interviewed with a semi-structured interview set. Semi-structured interviews were employed in order to allow producers to describe heirloom corn and its biocultural heritage however necessary. The interviews were recorded, transcribed, and translated for analysis using codes that identify the scope of publicness of colored heirloom corn. Interviews were transcribed and independently analyzed to ensure unbiased interpretation of the data. Field notes were gathered and analyzed from interviews and observations.

The first key actor interviewed coordinates the production of 30 producers in the region who plant and harvest colored corn. This variety of corn is produced individually, as a single crop or together with creole or hybrid white corn and other crops such as beans and squash within the milpa as a polyculture. According to information from Conabio, in the Ixtenco, Tlaxcala area, there are 40 native corn producers [20]. The first key actor shared the contact information of other producers with whom three other interviews were conducted using the snowball technique. The producers interviewed are part of the first producer's organization, but they also have a connection with other individual producers.

The authors used the constant comparison method, where the primary and secondary case study data was compared at the same time that new data was gathered and interpreted within the context of the theories proposed. The data obtained allowed us to collect the necessary information for analyzing the chain with the indicators proposed in Table 1. With the theoretical framework and methods and materials in place, we now proceed to explain heirloom corn's history and context in light of key concepts like territory, biodiversity, and community-based enterprise. We then develop the case study to understand the relevance of this value chain.

## 4. History and Context for Heirloom Corn

### 4.1. Territory and Biodiversity

Frequently linked to a territory, the concept of cultural biodiversity implies a sense of belonging to a social group characterized by certain cultural traits, including customs, values, and beliefs [21]. Although territory occupies a central place, as a dynamic element, from the eco-development and human ecology perspective [20] it goes further to integrate with cultural identity and sustainable development. Although multicultural by nature, Mexico's development has been defined by a hegemonic Western perspective that discriminates against indigenous or rural displays, such as—for example—the "milpa" [22].

The "milpa" is a Mesoamerican agricultural system of polycultures that is not only the pillar of the rural Mexican diet, but is also a common good at the heart of diverse regions' biocultural heritage [23]. It is "a space where cultures are recreated, autonomy is built, biodiversity is conserved, human rights are satisfied and food sovereignty is fostered" [22]. Heirloom corn is the milpa's central crop [20,24], and one of the most important tools for combating climate change [23]. Likewise, it is an emblem of Mexican cultural identity that goes back to native inhabitants: "Our grandparents wrote in the Popol Vuh that we are men and women of corn" [22]. For these peoples, corn signifies life and "a symbiotic relationship between man and corn [exists]: without corn, man would not exist and if man did not sow it, corn would disappear" [25]. That is one of the main reasons why the Puebla Plan set out to promote hybrid corn varieties at the end of the 1960s.

However, the agrarian policies associated with the Mexican Revolution for rural development contributed to the modernization of rural sectors, through—for example—the use of tractors, fertilizers, and improved varieties [1]. Despite these policies significantly replacing heirloom corn [26,27], producers still defend it as an icon of cultural identity. Of course, growing heirloom corn represents an opportunity at differentiation in the area of basic grains, however, producers' motivations go beyond yields or the modernization/mechanization of agriculture, as evidenced in the movement "*Sin maíz no hay país*" (without corn, there is no country), which aims to highlight the cultural heritage of this regional good [1].

The development of territorial products with cultural identity strengthens rural social groups' capacities [28] which helps revive rural areas by improving the quality of community life as a result of economic activities related to said goods, be it the sale of products at market or activities related to tourism [29]. Ixtenco, a small town in the rural state of Tlaxcala, is a clear example of this. There, heirloom corn conjures up the cultural identity of the producers who live there, who consider themselves "guardians" of that cultural heritage, which goes back many generations, and which they want to preserve and enhance [30]. The festival that celebrates the town's patron saint, Saint John, every June revives that tradition, mobilizing local actors that contribute to its organization, increasing social capital, and strengthening community identity by way of their connection to a product—in this case corn—and a territory, namely Ixtenco, Tlaxcala.

Biocultural heritage is understood as an "asset" constituted by a territory's natural and cultural resources. This is so insofar as the populations that manage them do not do so for exclusively economic ends; they also seek to strengthen identity, the local social fabric, as well as engage in environmentally respectful practices [31]. They do so by means of their knowledge systems, forms of territorial management, as well as their multiple under-

takings (fishing, agriculture, silvopastoralism, forest management, gastronomy, tourism, handicrafts, and artistic activities, among others). In this sense, heirloom corn connects the three factors that make up biocultural heritage, biodiversity, goods and services, as well as regional culture.

As the center of the milpa, corn supports other basic products, such as squash, chili, and beans that together create an agro-biodiverse system that shields the whole from pests, disease, and the effects of extreme climates. Likewise, it supplies the earth with necessary nutrients and allows other species to grow. In cultural terms, corn has been part of the history of Mexico for about 9000 years, although it began to appear in Mesoamerican iconography starting about 3000 years ago in the Olmec culture [25]. That history implies a traditional sowing and harvesting process, and—above all—the preservation of native corn seeds, including colored corn.

### 4.2. Community, Entrepreneurship and Development

The biocultural approach herein described defies the mainstream economic theory, built on the assumption that business's ultimate goal is profitability and to maximize wealth-creation [32,33]. However, it has been empirically demonstrated that entrepreneurship can seek other values and goals beyond profitability, including autonomy and innovation [34]. In an attempt to expand this concept, Rindova, Barry, and Ketchen define entrepreneurship as the "efforts to bring about new economic, social, institutional, and cultural environments through the actions of an individual or group of individuals," focusing that it is a social change activity with a variety of possible outcomes—including economic gains—instead of an economic activity with social impact [35].

The concept of Community-Based Enterprise (henceforth CBE) [36,37] has emerged as a theoretical construct that integrates these different dimensions of entrepreneurship (economic, social, political, cultural, and environmental) as a strategy for sustainable local development in poor populations. A CBE can be defined as "a community acting corporately as both entrepreneur and enterprise in pursuit of community common good" [36]. The community is defined by geographical proximity and a collective culture or ethnicity that acts as a relational bond. As an entrepreneur, its members "collaboratively create or identify a market opportunity, and organize themselves in order to respond to it" [36]; as an enterprise, they "work together to jointly produce and exchange goods and/or services using the existing social structure of the community as a means of organizing those activities" [36]. It is important to note that the CBE should be owned and managed by its members, who—although with different levels of commitment—act corporately. In a CBE, profit-making is not the primary goal but rather a means to make the activity sustainable at the sake of other community purposes, including the desire to control their own local development [35]. Indeed, a multiplicity of goals, together with the integration of ancestral and new community skills, as well as community participation in the form of cooperative practices and values are the main characteristics of CBEs [37,38]. Among other factors that explains the emergence of CBEs, Peredo [37] highlights: a lack of individual opportunities, processes of social and cultural disintegration, and environmental degradation.

CBEs emerge in "community-oriented" societies [39], whose members share a community commitment to the cultural context within their locality. This community orientation—expressed in social networks [40] of trust, solidarity, and reciprocity—builds social capital, defined as "the aggregate of the actual or potential resources which are linked to possession of a durable network or more or less institutionalized relationships of mutual acquaintance and recognition" [41]. Social capital binds the members of a community together [42] and, during social and economic crisis, its presence activates nets of solidarity that help them to better face those circumstances by sharing goods (and risks).

The anthropological basis of this reciprocal co-operation goes against the individualism that characterizes mainstream economics, and it is at the heart of the so-called "logic of gift" [43,44]. In a "gift-economy", members of the community frequently exchange goods or services on a regular basis, without any explicit agreement of the "*quid pro quo*"

that characterizes the commutative justice of markets [36], but rather aimed at creating strong mutual relationships beyond the logics of duty and self-interest [45–47]. From an inter-generational lens, what is received from previous generations is an incommensurable wealth which cannot be repaid but rather honored (it is priceless). As Enderle [48] describes when speaking about public goods,

> Production of public goods is based on human relatedness and needs other-regarding motivations such as gratitude for the gifts received, entrepreneurial spirit and service to others. Commitment to public goods does not earn immediate rewards, may offer uncertain personal benefits in the future and can even demand personal sacrifices. But, not infrequently, it is actually made because the interests of other people count, their rights are to be respected, and the needs of the community and society should be addressed.

This is evident in a set of cultural values that are expressed in some kind of goods, such as the case of corn in Mexico. Corn has been a central element in the Mexican culture since ancient times, and the recovery of different varietals can not only become a competitive advantage but can also contribute to this common sense of purpose; "What is striking, however, is the way that in CBEs this 'novelty' is a legacy of long-standing resources in tradition and culture" [36]. Moreover, in socioeconomic terms, corn is one of the most consumed foods in the country—among other aspects—due to the nutritional characteristics it contains, as we will show in the following section.

## 5. Results and Discussion

In Mexico, the main crops for consumption that have value in terms of territorial cultural heritage are, firstly, corn, and secondly, coffee [49]. This territorial element is based on production, given that most of the producers in the country consider themselves small-scale producers, in addition to farmhands. Producers and farmhands are often in situations of marginalization in terms of poverty level, lack of basic services, and Human Development Index numbers lower than the national average—some of which are comparable with countries considered the poorest in the world. There are, of course, some exceptions, as seems to be the case in Tlaxcala (as we will see below in the Economic Geography section). In addition to this, many are part of indigenous communities, which are associated with social differentiation and limitations to improvement of the environment in which they operate. At the same time, within the sphere of consumption, the production of these crops is vital because it satisfies the basic needs of those who produce it, either through self-consumption or through exchange for other things, including foods like poultry, milk, vegetables, or fruits.

Given that this agro-food sector (basic grains) has been among the most highly impacted by crises over the last 30 years and, especially, by the liberalization of the market in 1994 through trade agreements, which spurred the closure of institutions that truly protect the interests of small-scale producers, farmers have sought alternatives to meet their needs. A recent one includes differentiating their products based on specific characteristics associated with the origin of production. Following this, producers become interested in offering different corn with extraordinary qualities that are more valued; vendors become interested in promoting a traditional product that can be classified as gourmet, and consumers respond by eating a product with a familiar, but different flavor in a responsible and sustainable way. These factors combined make territorial appreciation of a locality possible.

Applying the methodology proposed in Table 1, each dimension of the chain is analyzed in order to evaluate the good's level of publicness.

### 5.1. Input-Product Dimension

#### 5.1.1. Characteristics of the Product on the Market

Turrent and Serratos [50] define heirloom corn as the set of the grain's regional varieties in Mexico, which continue to evolve, change, and emerge. Mexico produces 64 of the 220

breeds of corn found in Latin America, of which 92% are considered native [20]. Each race can present different varieties and be combined to give a wide range of colors and other specific characteristics particular to each region in Mexico. Due to their elevated production volume, the white grain for human consumption, fodder grain for animals and yellow grain stand out; in terms of preservation and tradition, blue corn and colored grain stand out. In Tlaxcala, there are 12 primary corn races, among which *Chalqueño*, *Cacahuacintle*, *Conical and Elotes Conical* (blue, black, red) stand out, as well as *Chalqueño-Bolita* (purple), *Chalqueño-Cacahuacintle*, *Chalqueño-Conical* (white, red cob), *Conical-Elotes* (blue), *Conical-Cacahuacintle*, *Conical-Bolita* (purple), *Conical-Chalqueño* and *Conical-Pepitilla* [51].

The tortilla is the main product derived from heirloom corn, although it is also used for 600 forms of food that are only produced nationally. A June 2020 study from the Center for Food and Development Research (CIAD, for its initials in Spanish) and the National Autonomous University of Mexico identified the nutritional characteristics of blue corn, which can be identified in at least 50 races [52]. The results reveal that heirloom blue corn tortillas contain more antioxidant, anti-inflammatory, and anticarcinogenic properties than industrial tortillas sold in supermarkets. Therefore, included in a balanced diet, native corn is not only differentiated based on physical qualities that are perceived with sight and taste, but also through its nutritional properties. Tlaxcala's varieties comply with the differentiation of the product in terms of the corn's colors, its nutritional properties, and the preservation of these species.

### 5.1.2. Income Distribution along the Chain

According to spring–summer 2020 estimates from the Bank of Mexico in collaboration with the Trust Funds for Rural Development (FIRA, for its initials in Spanish), the expected yield per hectare is 4.5 tons, while the price is US $225 per ton. Thus, the total income adds up to an approximate of US $1012.5 per hectare. The costs amount to $896, so the producer expects a profit of US $116.5 per hectare with a unit price of US $199 per ton. That is to say, the producer can obtain an 11.54% profit per harvested hectare [53]. Producers have costs that can be financed by the government (fertilization and cultural labor), mixed costs (planting and pest, weed and disease control), and non-financial costs (preparing the ground, harvesting, selection and packaging and other costs like storage and commercialization) that they must absorb individually. The latter can decrease through the economies of scale generated by being organized. Participation in these activities significantly reduces costs, which collectively benefits associated producers.

### 5.1.3. Chain Structures for White and Heirloom Corn

The main activities carried out in any corn chain, whether white or heirloom, can be summarized in supply, production, storage, transformation, marketing, and consumption. From a static and linear point of view, each link is the responsibility of an economic actor or agent and one activity leads to another. However, as presented by Ayala, Quirós, and Saravia [54], analysis must be done in the environment under a social dynamic that includes the specific production aspects pertinent to each grain chain [54]. According to the study carried out, the chain is different from previous ones since three intermediate links are eliminated, which benefits the producer with a greater percentage of the final added value. The current chain of heirloom corn in Ixtenco, Tlaxcala is simple with only two agents: the producer (organized small-scale producers), who sows, harvests, works and nurtures the land, and the vendor, who transforms, collects, markets, and sells the final product as food (e.g., tortilla, dishes, kernels). In the case of heirloom corn, producers' families—especially their mothers and grandmothers—preserved the original seeds, which are planted to market these races as a gourmet product. With this legacy, the producers plant the seeds, new ones are collected at harvest, which are then planted, and so on, that is, they do not need to buy seed from an outside source. At the same time, the vender proposes seeds and fertilizers that are less harmful to the harvesting patch, which corn research institutes in Mexico have developed. In this way, there is a short chain in the sense of elimination of

intermediaries, which benefits the producer who receives greater added value from the final product; lasting relationships and trust are created between the agents, and risks are distributed between them. However, the producers' total dependence on the vendor could be harmful if the latter loses interest or goes out of business. It is a risk that both decide to take, but it undoubtedly affects the producer more because they would lose at least one year of harvest trying to find another buyer under the same conditions. This situation makes visible the vulnerability of this potential public good since it is undervalued and a crisis can set it back so significantly.

*5.2. Economic Geography*

The production of colored heirloom corn is marginal with respect to the production of white corn for human or animal consumption. Production does not usually surpass 10 tons per producer, which corresponds to what the sole marketer is able to buy. These small-scale producers come together to sow on various fields and/or to rent land and obtain a greater volume of production. On average, they have less than one hectare to produce on, which, as noted, yields about 4.5 tons per hectare in the study area.

Producers use the milpa system in which heirloom corn (different species), beans, chili and other corn, such as white and hybrid (genetically modified), coexist. Therefore, while they could obtain 4.5 tons, the system they use yields less in order to preserve a healthy coexistence. The producers share socioeconomic characteristics with the majority of producers in Mexico's agro-food system. They are small-scale producers living in marginalized conditions, often lacking basic services like sewage and gas; their houses lack concrete flooring [55]. However, the region's Human Development Index was high in 2010 and 2015, around 0.732 and 0.742 respectively, since education, health, and income have remained stable thanks to producers' collective organization, who have maintained an increased rate of production and have promoted initiatives in these areas [56].

In terms of consumption, they operate within a short value chain based on the absence of an intermediary. The buyer buys directly from the producer (or producer group) and sells to the final consumer. This chain is unique based on the fact the buyer is a spokesperson for the importance of heirloom corn, thus they are the one interested in the product's revalorization in the market. This actor obtains the heirloom corn, nixtamalizes (processes) it and then makes native or heirloom corn tortillas, which is their main product for sale. They sell this product in shops in Mexico and also distribute it to restaurants. It is important to mention the connection between the producers and the buyer since both generated profits for a period of time, although they did not manage to break even due to the current pandemic.

In addition, it is worth mentioning the relationship between the producer and the final consumer that is established thanks to the marketer, who promotes this relationship. The final consumer is an actor outside the locality where the corn is grown, but is still in proximity to the area, has an urban profile with an above-average income level and is thus willing to risk buying a new product if it guarantees quality and differentiation over other existing products. In the case study, consumers and the marketer have established a relationship of trust and friendship that is expressed in the purchase of heirloom corn tortillas. At present, producers, marketers, and final consumers are going through a crisis due to the COVID-19 pandemic, so it has been difficult to maintain the product's value. In this way, this under-protected potential public good's vulnerability is present not only in the chain's structure, but also impacts economic behavior.

*5.3. Institutional Framework*

The protection of heirloom corn in Mexico is based on various international tools and national laws. Among them, the Aichi Targets stand out, whose strategic objective is, according to its webpage [57], "to improve the situation of biological diversity to safeguard ecosystems, species and genetic diversity." In addition, the Nagoya Protocol focuses on access to genetic resources and fair and equitable profit sharing that arises from their use

in the Convention on Biological Diversity. The Cartagena Protocol on biosafety of the Convention on Biological Diversity [58] is also important. At the national level, Mexico has a Biosafety Law for Genetically Modified Organisms (LBOGM) and a Federal Law for the Promotion and Protection of Heirloom Corn, which was recently approved on 13 April 2020.

Programs do exist, but they are neither sufficient nor efficient since they focus on economic support for harvest, pass through many intermediaries or take too long to arrive (e.g., after the harvest). Moreover, these economic supports lack continuity over time: government changes imply different public policies. Over the years, distrust or dependence have sprung up between producers and institutional actors.

Contradictorily, although international and national laws exist for the protection of the product, there is a total disconnection with the productive sector. Interviews with producers reveal that they prefer to continue the processes they have built socially; that is the cooperatives and societies that they have organized in recent years respond more to their own needs than to institutional programs. However, although these groups are more structured, are designed for the long term and include participation from several producers, they have not been sufficient for maintaining long-term appreciation of their product.

Regarding food safety, in Mexico's most rural communities, where heirloom corn is grown, food insecurity and hunger persist at rates documented at over 40% [59]. For many producers, who engage in what is called subsistence farming, farming staves off or reduces food insecurity and provides a significant amount of nutrients for household diets [60]. In the case of subsistence farming, heirloom corn is used to feed a household and the excess is sold as a public good.

The composition of a corn kernel contributes to human nutrition [61]. It contains many micronutrients, including vitamins, minerals, and phytochemicals essential for health. Populations that depend on corn as a staple particularly benefit from these nutrients. For example, the B-complex vitamins in maize are good for skin, hair, the heart, the brain, and proper digestion. The presence of vitamins A, C, and K, together with beta-carotene and selenium, helps to improve thyroid and immune system functioning. Potassium is a major nutrient present in corn and has diuretic properties. The presence of essential fatty acids, especially linoleic acid in maize oil, plays an important role in the diet by maintaining blood pressure and regulating blood cholesterol levels. Phytochemicals also contribute to antioxidant capacity [61].

Nixtamalization is a process by which heirloom corn is prepared for consumption. Therein, the corn is soaked and cooked in an alkaline solution, such as lime or, more traditionally, ashes, washed, and then hulled [62]. For its part, commercial corn does not require nixtamalization in order to be consumed. This process removes the pericarp from the grain and makes it easier to grind, increases flavor and aroma, and, importantly, increases key nutrient values. Nixtamalization has been very important to civilizations over time because, in the process, niacin is converted to free niacin, or B3, making it available for absorption into the body, thus helping to prevent diseases caused by amino acid deficiencies like pellagra. Masa, or corn dough, is formed from grinding the nixtamalized corn and provides an edible product for subsistence or for purchase on the market.

Food safety is linked to the publicness of corn in Mexico because, in addition to being a part of biocultural heritage due to its sociocultural content, it is Mexico's premier staple food. Corn represents half of the total volume of food consumed each year and provides the population with about half of required calories [20]. Colored corn is not only produced for self-consumption, but also to be exchanged for other essential goods when marketing it. It then benefits the consumer in terms of the provision of a public good that is also nutritious.

*5.4. Governance*

The governance associated with this chain pertains to a local one in which the buyer dominates since they have the power to negotiate the price per ton. A dependent rela-

tionship exists between the producer and the buyer characterized by a captive nexus that endangers the producer by having a sole buyer. In terms of conventions, there is a strong civic component in terms of responsibility for agrobiodiversity and an aspirational one insofar as the product is a gourmet corn variety that meets the quality standards of the end consumer.

Given these characteristics, governance can be classified within the G4A typology; there, heirloom corn producers sell to a single buyer without intermediaries, which includes a high level of coordination and of asymmetry. Producers have little independence and socialize the benefits, but not the risks, which they assume to a greater extent. This structure has been reinforced during the COVID-19 pandemic.

Due to the COVID-19 pandemic, producers reported having planted colored heirloom corn for the buyer, who, given the pandemic, could not buy the harvest. To avoid totally losing the harvest, the corn they grew according to the buyer's requirements was ultimately sold at the price of white corn for animal consumption. This is the danger of having only one buyer. It is worth mentioning that the local producers approach the situation with an abundance of resilience; they emerge from each crisis and resolve problems at each step in order to move forward with the production process and, beyond that, with the preservation of this type of corn. Thus, it is all the more important to value heirloom corn production as a potential public good.

## 6. Conclusions

This analysis of the heirloom colored corn chain reveals its status as a potential public good. Of note, within the locality studied, the appropriation of this good is communal to the producers. Any corn producer can produce heirloom colored corn, which is why colored corn is considered to be a potential public good on a country level.

As one of the main foods in Mexico, corn is preserved and conserved by actors at different levels. In recent times, local actors in the rural productive space are in charge of preserving colored corn throughout the generations, while external actors, identified as traders, promote final consumption in the cities. At the institutional level, four drivers protect this asset as a traditional food. In the first place, universities and research institutes conduct technical, economic, sociological, and anthropological studies on native corn and corn in general. Second, the government at the local and federal levels, as well as decentralized public organizations such as CONABIO, propose projects, plans, and strategies for the preservation and conservation of heirloom corn, as well as provide economic and in-kind support for producers. Third, there are non-governmental, national, and foreign organizations, associations, groups, and cooperatives that are not local producers per se, but encourage the preservation and conservation of native corn. Finally, there are international organizations that have an interest in analyzing this chain, such as UN Environment and CYMMIT.

The region's native colored corn chain shows signs of vulnerability as an agri-food commodity since it was previously under-exploited and only now, through the producer organization's initiative and the buyer's interest, has begun to recover. However, the pandemic and attending crisis has quickly revealed heirloom corn's fragility and weak points, including especially the extreme dependence between the only two actors in the chain.

Another visible problem includes the disarticulation between the different actors and, mainly, between local and external actors. Both are interested in preserving, conserving, and promoting native corn, but their proposals and initiatives fail to link up. The current government is expected to reignite confidence in institutions in order to achieve long-term sustainable local development. Likewise, ties between producers and other economic agents such as consumers must be strengthened. In this sense, the organization itself is trying to reach the final consumer to educate about the territory and the importance of this variety of corn.

The Ixtenco producer organization's initiative has been replicated in other nearby regions like Puebla, Mexico, Querétaro, and Guanajuato. We thus expect to reproduce and corroborate this analysis with producer organizations in these states.

As noted, corn is the center of the milpa system, which is why it impacts other agro-food goods, such as beans and chili, in addition to promoting agrobiodiversity. Likewise, producers within the community have begun to organize, which allows for commercialization on a larger scale, providing employment and generating income, at the same time that it generates social cohesion. Likewise, the territory already has a certain reputation for the production of heirloom corn, although obtaining a GI has not been possible. And even though the value chain has been defined, and conservation and preservation laws are in place, formal protection of this good still lacks.

In terms of publicness, heirloom corn is a second-tier good, although it has the potential to be classified as part of biocultural heritage. To achieve this, two major challenges remain: the first involves addressing the main weakness of the overall Mexican agro-food system, namely formalizing relationships among formal internal and external actors at the institutional level. Such a step would help strengthen trust and formalize production networks. The second challenge involves keeping pace with production and marketing over the long term, which can be achieved through awareness on the part of end consumers. That is, a greater number of buyers must become aware of the benefits of heirloom corn, a task that the sole marketer had been doing to date, but which the Covid-19 pandemic has brought to a halt.

In the long term, a development strategy is needed that, in addition to linking actors and resources, involves the public sector and further expansion of the private sector (buyers). This strategy requires the development of a proposal that includes formal collective rules, which would draw from the cooperative, but would be written and accepted by the actors involved. This would guarantee the long-term sustainability of corn production and, in addition, would help make it part of biocultural heritage, generating economic development in the community as a local product rather than as a GI.

The publicness of heirloom corn in the area we studied clearly impacts biocultural heritage. However, that impact has not been fully appreciated because participation from a strategic actor within the local government as a formal institution is lacking. Although this issue has been studied since the 1990s, the institutional apparatus must be discussed since local actors rarely have access to this information. Therefore, part of the attending strategy must include socialization of knowledge in given areas.

**Author Contributions:** Conceptualization, M.V.-S., G.S., C.B.S.; methodology, M.V.-S., G.S., C.B.S.; data collection M.V.-S., G.S.; formal analysis, M.V.-S., G.S., C.B.S.; writing—original draft preparation, M.V.-S., G.S.; writing—review and editing, M.V.-S., G.S., C.B.S. All authors have read and agreed to the published version of the manuscript.

**Funding:** This research received no external funding.

**Institutional Review Board Statement:** Not available.

**Informed Consent Statement:** Informed consent was obtained from all subjects involved in the study.

**Data Availability Statement:** Data is available to review by request from MVS.

**Conflicts of Interest:** The authors declare no conflict of interest.

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
