# Peer review of "Colored Heirloom Corn as a Public Good: The Case of Tlaxcala, Mexico"

_sustainability, doi:10.3390/su13031507_

Round 1
Reviewer 1 Report
Referee Report for Sustainability -1037571
Colored Heirloom Corn as a Public Good: The Case in Tlaxcala, Mexico
This study “analyzes the case of colored heirloom corn in Tlaxcala, Mexico as a public good associated with the region’s biocultural heritage”. Although the topic under study could be interesting, and the general idea of Geographical Indications, Public Goods and Sustainable Development deserves attention, I am afraid that the conclusions of this study are not relevant enough to be published in the Sustainability journal as the innovative character of the research and its contribution to the advance to the state-of-the art are not evident. While theoretical foundations are reviewed its approach is not sound enough and lacks rigor. Moreover, the research methodology and approach used by the authors are not convincingly described, as the paper lacks a Methodology section, with proper justification for the choices of methods for data collection and analysis (instruments used to collect data? Primary and secondary data?...). Also, result analysis and discussion are weak and conclusions are vague. The authors conclude “This analysis of the heirloom corn chain reveals its status as a public good valued intermittently (line 509)”, and this conclusion is reached considering the Tlaxcala case study only. However, they also recognize “heirloom corn is not strictly unique to their region” (line 547), which means that no generalization can be made for this product in Mexico based just on a single case study. Also, the authors conclude that “Heirloom colored corn does not have GI and no proposal for its specification is pending (line 546)… A DO label is possible for one of the good’s byproducts, i.e., colored heirloom tortillas, although such a DO would have to include the entire country, and not just the state of Tlaxcala (line 549- 550)”, which further reinforces the concerns about the relevance of this study. Focus and policy implications of this research should be clarified.
Some additional comments:
- in its current form, the paper is difficult to read as it goes too length in several parts, especially in the “Theoretical Framework” and “History and Context for Heirloom Corn” sections. These sections are too long, sometimes repetitive and should clearly be more focused on the areas the paper is intended to contribute. For example, “II. History and Context for Heirloom Corn” section could be significantly reduced, its subsections deleted, and the main ideas summarized. Also, almost 2 pages are dedicated to a “Food Security” section, with details about the composition of corn kernel (in terms of proteins, vitamins, etc.) that were irrelevant for the authors’ study. Two or three sentences from this section would be enough to highlight the importance and properties of the product, as well as to explain the difference of the process for the preparation of heirloom corn and “commercial” corn.
- the authors use interchangeably and not in a coherent way the terms “public goods” and “common goods”, which are not the same thing for the Economic science. Although Samuelson (1994, 1995) are cited to define public goods and its characteristics, the authors should use additional references to clarify the concept of public goods used. The authors follow Belletti et al (2017) approach but lack the rigor of those authors. As an example of a clear and summarized approach for the classification of public goods, for example, the work of Arfini et al (2019, “Are Geographical Indication Products Fostering Public Goods? Some Evidence from Europe”, published in the Sustainability journal.
- Citations / references should support statements like “This theoretical framework has been used for several years, dating back to its proposal in the 1990s”, for example.
- On line 364, of the 14th page of the paper, the authors say: “Heirloom corn refers to corn that is considered part of native corn crops, whose origin dates back 6,500 years with teocintle which it is thought to be the parent plant of corn and which is still produced”. Shouldn’t this information be presented be at the beginning of the paper, when first referring to this culture?
- On Value Chain section (pg. 16) the authors refer they use the Ayala, Quirós & Saravia, (2019; 133-134) “theoretical basis for presenting the structure of heirloom corn in the region under analysis” whereas in the Introduction section it’s only mentioned that the “corn chain is explained using the Gereffi’s Global Value Chains approach (1994, 2005)”
- “According to the study carried out” (line 416, pg. 17)… how was the study carried out is a question without clear answer in the paper.
- On pg 18 the authors state: “The producers share socioeconomic characteristics with the majority of producers in Mexico’s agrofood system. They are small-scale producers living in marginalized conditions, often lacking basic services like sewage and gas; their houses lack concrete flooring. In terms of poverty measurement, they align with some of the Human Development Index and Coneval Index figures”. This characterization should be supported with official data (presented in a table, for instance). HDI is mentioned again (in a previous section was said this producers had a lower HDI than average).
- Also, on lines 456-458 the authors say: “It is important to mention the connection between the producers and the buyer since both generated profits for a period of time, although they did not manage to break even due to the current pandemic.” What is the source for this information? What producers? How many? Is this their perception (registered on an interview) or did you have access to financial data of these producers before and during the pandemic crisis?
- Line 464- 465: “At present, producers, marketers and final consumers are going through a crisis, so it has been difficult to maintain the product’s value.” How do the authors support this sentence? What’s the data? What was the product’s value before and after the crisis?
- Line 477: what does the authors mean by: “arrive at the wrong time”?
- Line 477: what does the authors mean by: “Total discontinuity exists…”
- Line 482: “Interviews with producers reveal that they have no interest in claiming any benefit from the government and prefer to continue doing what they have built socially.” Finally an explicit reference is made to the methodology used by the authors! However, the number of producers interviewed, the method employed, how was it registered, if it was a structured interview or not, etc.)
- Line 503-505: “It is worth mentioning that the local producers approach the situation with an abundance of resilience; they emerge from each crisis and resolve problems at each step in order to move forward with the production process and, beyond that, with the preservation of this type of corn.” What are the grounds for this statement? It seems the authors’ opinion, which it is not supposed in this type of work.
- On line 514 the authors mention: “As a third characteristic…” but before do not clarify what are the other two.
Reviewer 2 Report
The article requires some refinement in terms of key methodological elements. The purpose of the research was not clearly defined. There is no ‘Materials and methods’ section, in which the applied methods would have been described and the object of research would have been characterized in detail.
The analyzed case is interesting, the conclusions from the literature review are properly presented, the theoretical framework of the research is described in detail, but due to the disturbed structure of the paper, it is difficult to assess the subject and scope of the research. There is also no results discussion containing references to the results obtained by other authors.
The article structure is not tailored to the technical requirements of the journal. The division into the required sections is missing, the reference pattern in the text is incorrect.
Minor remarks:
There is no reference in the text to Table 1.
In Table 2 - there is an error in the word ‘governance’.
Reviewer 3 Report
The article entitled "Colored Heirloom Corn as a Public Good: The Case in Tlaxcala, Mexico" attempts to establish the case for coloured heirloom corn as part of the biocultural heritage that can be associated as a force for public good. This is of interest especially in relation to biodiversity and food sovereignty. However, the paper lack coherency and appears to be a string of ideas that are badly connected.
I will suggest that the authors seek a professional for the English language editing. In most cases, the key messages are not clearly articulated, the sentences are not clearly connected o keep the flow for interested readers.
A few suggestions:
- The Abstract need to revised to capture the key messages of the article
- L22, L26: grammartical error
- L97-99: I suggest a max. of four references cited here
- L125. Table 2 "Governance" instead of "Gobernance". The table can made more visible.
- L132-133: Max. of four references
- L159-160: name in full first followed by the abbreviation, e.g. Food and Agriculture Organization of the United Nations (FAO)...
- L166-167: revise for clarity
- L205-206: At the end of the 1960s, the Puebla Plan set out to promote hybrid corn varieties. The sentence seems incomplete?
- L213: revise for clarity
- L218: ..themselves as 'guardians'
- L223-228: The sentence is quite long, biocultural heritage should have been defined much earlier in the article e.g at the introductory stage.
- L251: something missing after environmental--
- L269: ....within their location.
- L303-304: The sentence need to be revised
- L310: a more recent can be cited here
- L332: ....in order to be consumed.
Round 2
Reviewer 2 Report
The Authors responded to all my comments. The new version of the manuscript was significantly improved, and the changes make it clear and understandable for readers. I only have doubts whether changing ‘Conclusions’ to ‘Discussion’ is appropriate. I would propose to stay with the section ‘Conclusions’ as in the first version of the article and add discussion to the section “Results”, because in fact it contains some references to other research results.
Author Response
Dear Reviewer 2:
Thank you very much for your positive evaluation of our second version, as well as your recommendations to continue improving the manuscript. As you suggested, we replaced “Discussion” with “Conclusions” and made the arguments and discussion surrounding the findings more coherent and balanced. A professional proof-reader has also edited it, as suggested, and all the remaining technical aspects have been addressed, especially the reference system.
We are very grateful for your feedback that have improved our draft immensely.
Respectfully,
The authors
Reviewer 3 Report
The authors have made good progress with the paper. It is now well structured and much improved from the previous version.
Author Response
Dear Reviewer 3:
Thank you for your new revision of our paper, “Colored Heirloom Corn as a Public Good: The Case of Tlaxcala, Mexico.” We have made considerable effort to improve this new draft, reviewing the methodological basis, as well as making the arguments more balanced and coherent. A professional proof-reader has also edited it and all the remaining technical aspects have been addressed, especially the reference system.
Again, we are very grateful for your comments to improve our work and we hope all of them have been fully addressed.
Respectfully,
The authors